# How Can a Bundled Payment Model Incentivize the Transition from Single-Disease Management to Person-Centred and Integrated Care for Chronic Diseases in the Netherlands?

**DOI:** 10.3390/ijerph20053857

**Published:** 2023-02-21

**Authors:** Sterre S. Bour, Lena H. A. Raaijmakers, Erik W. M. A. Bischoff, Lucas M. A. Goossens, Maureen P. M. H. Rutten-van Mölken

**Affiliations:** 1Erasmus School of Health Policy and Management, Erasmus University Rotterdam, 3062 PA Rotterdam, The Netherlands; 2Department of Primary and Community Care, Radboud Institute for Health Sciences, Radboud University Medical Center, 6525 GA Nijmegen, The Netherlands; 3Erasmus Choice Modelling Centre, Erasmus University Rotterdam, 3062 PA Rotterdam, The Netherlands; 4Institute for Medical Technology Assessment, Erasmus University Rotterdam, 3062 PA Rotterdam, The Netherlands

**Keywords:** bundled payment, integrated care, person-centred care, payment model, chronic diseases, multimorbidity, primary care, value-based payment

## Abstract

To stimulate the integration of chronic care across disciplines, the Netherlands has implemented single-disease management programmes (SDMPs) in primary care since 2010; for example, for COPD, type 2 diabetes mellitus, and cardiovascular diseases. These disease-specific chronic care programmes are funded by bundled payments. For chronically ill patients with multimorbidity or with problems in other domains of health, this approach was shown to be less fit for purpose. As a result, we are currently witnessing several initiatives to broaden the scope of these programmes, aiming to provide truly person-centred integrated care (PC-IC). This raises the question if it is possible to design a payment model that would support this transition. We present an alternative payment model that combines a person-centred bundled payment with a shared savings model and pay-for-performance elements. Based on theoretical reasoning and results of previous evaluation studies, we expect the proposed payment model to stimulate integration of person-centred care between primary healthcare providers, secondary healthcare providers, and the social care domain. We also expect it to incentivise cost-conscious provider-behaviour, while safeguarding the quality of care, provided that adequate risk-mitigating actions, such as case-mix adjustment and cost-capping, are taken.

## 1. Introduction

In many countries, the prevalence of chronic diseases, and in particular people with multimorbidity, i.e., two or more chronic diseases, is increasing [1]. Two thirds of people over 45 will develop multimorbidity in their remaining lifetime [2]. To address their needs, many countries are now implementing different models of integrated care [3]. As the Netherlands were among the first countries to do so on a very large scale, there are lessons to be learned for other countries from how this evolved in the Netherlands, in particular with regards to possible incentives for truly person-centred and integrated care. 

Historically, the Dutch healthcare system has had a strong primary care sector, in which general practitioners (GPs) act as gatekeepers to secondary care (i.e., patients need a referral by the GP) [4]. To improve the quality of care to people with chronic diseases, single-disease management programmes (SDMPs) have been introduced in Dutch primary care since 2010, for diabetes type-2 (DM2) [5], cardiovascular risk management (CVR) [6], and chronic obstructive pulmonary disease (COPD) [7]. Therefore, the GP is the main caregiver for many patients with chronic diseases in the Netherlands. These SDMPs were based on chronic care standards, which are essentially clinical guidelines for providing high quality, multidisciplinary, integrated care. 

To coordinate the implementation of the SDMPs in a region, a new organisational entity, the primary care cooperative (care group), was introduced. Today, there are 130 primary care cooperatives in the Netherlands based on the collaborations of general practices [8]. For the daily execution of the SDMP’s and to reduce the workload of GPs, a new professional role was introduced in the GP practice, namely that of the nurse practitioner. The nurse practitioner regularly monitors symptoms and physiological parameters of patients with the chronic diseases mentioned above, and provides lifestyle and coping advice [9].

To further incentivise the integration of multidisciplinary care, the implementation of the SDMPs was supported by a bundled payment model [10]. The bundled payment covers the costs of coordination, the costs of regular check-ups by the nurse practitioner or the GP, three hours with the dietician for people with DM2, the foot therapist for patients with DM2, the physiotherapist for patients with more severe COPD, and a single (tele)consultation with a medical specialist when necessary. Health insurers contract primary care cooperatives, which in turn subcontract GPs and other healthcare providers for providing the services in the bundle [11]. The fee of the bundled payment results from the negotiation between the primary care cooperative and the health insurer about the content and price of the services in the bundle, which thus varies between primary care cooperatives. 

Compared to other countries, the scope of the bundled payment in the Netherlands is limited, both in terms of target population and services included in the bundle. For instance, in the United States, accountable care organisations are generally responsible for all healthcare expenditures of a delineated patient population [12,13]. In the Gesundes Kinzigtal programme in Germany, the target population includes a group of 33,000 patients from Baden-Württemberg, who are insured by two public health insurers. Key characteristics focus on prevention, self-management, reduction of polypharmacy, patient-centred care, and shared decision-making. The programme is funded by a capitation-based payment combined with a shared savings model [14,15]. In the United Kingdom (UK), general practices receive a lump sum for all GP-care, some specialist care, and generic medication [16,17]. In the UK, integrated care organisations are introduced to stimulate integration between primary care physicians and specialists. The integrated care organisations are responsible for a case-mix corrected budget per capita [18].

As a result of the introduction of the SDMPs in the Netherlands, the vast majority of patients with DM2, CVR, and COPD are now treated in primary care. The quality of chronic care is monitored by InEeN, a primary care interest organisation, which annually publishes process- and outcome-indicators at the care-group level [19]. These indicators were found to improve over time [20], but the clinical relevance and long-term impact of these improvements are uncertain [14,20,21]. Improvements in the work experience of GPs were also reported [9,14,20].

However, the SDMPs have several limitations. First, the chronic care programmes focus on a single chronic disease, rather than adopting a holistic approach that considers the social context of the chronically ill patient (e.g., family, living environment, financial resources, and the work situation) [21,22]. The programmes mainly aim to improve clinical disease-specific indicators, and there is less attention paid to psychological and social aspects. This does not match well with how our perspective on disease and health has evolved. In the Netherlands, many primary care cooperatives have recently embraced the new concept of so-called positive health (‘health as the ability to adapt and to self-manage, in the face of social, physical, and emotional challenges’) that was introduced in 2011 by Huber et al. [23,24]. Second, the scope of the services included in the current bundles is limited. The bundled payment does not cover care that transcends the chronic disease [25,26]. The bundled payment does not include all primary healthcare, no secondary care, no mental health care, and no social services. It might stimulate collaboration between healthcare providers in primary care (e.g., between the GP and the dietician), but less so between the GP and the specialist or between the GP and the social worker. 

The introduction of the SDMP and the bundled payments were expected to improve the efficiency of care delivery and reduce healthcare expenditures or the growth thereof [27]. However, there is evidence that they increased the total costs of healthcare, especially in patients with multimorbidity [14,28]. This cost increase probably results from a combination of the detection of unmet needs in patients with multimorbidity, double declarations, and an incentive to refer the more complex patients to secondary care to avoid costs exceeding the bundled payment [28]. The currently used SDMPs and bundled payments are not suitable for patients with multiple chronic diseases. 

As a result, we are currently witnessing several initiatives to broaden the scope of the SDMPs aiming to provide person-centred and integrated care (PC-IC) [28,29]. This raises the question of which payment model would best support this transition [29]. As a first step, InEeN proposed merging the current bundled payments for people with multiple of the respective chronic diseases to remove duplication [30]. However, that proposal would still not fully incentivise PC-IC. This paper aims to present an alternative payment model that incentivises the integrated nature of a PC-IC programme for people with chronic diseases. It is based on a targeted literature review of (incentives in) traditional and more recent payment models in different countries and inspired by a specific PC-IC initiative in the Netherlands. 

## 2. Methods

### 2.1. Case Example: OPTIMA FORMA

The proposed payment model was specifically designed to match with one of the initiatives to move towards PC-IC in the Netherlands, i.e., the project, OPTIMA FORMA—Towards a patient-centred multimorbidity approach for chronic disease management in primary care. In this project, healthcare providers, patients, GP experts with a special interest in DM2, COPD, or CVD, primary care cooperatives coordinators, and researchers developed a new integrated care programme that goes beyond the disease-specific clinical domain. The new care plan has a quadruple aim: (1) enhancing patient experience, (2) improving population health, (3) reducing costs, and (4) improving the work life of health care providers [31]. 

In the PC-IC programme, a holistic assessment of the health status is performed, personal goals are set, and interventions to achieve these goals are put in place [32,33]. The first step in this programme is assessing the integral health status of the patient (health across multiple domains—Figure 1), using a (preferably digital) questionnaire at home and physical measurements (i.e., blood pressure, weight, and glucose levels). The second step is an appointment in which the results are discussed with the patient in a semi-structured way. The case manager asks if the patient recognizes himself in the results of the assessment, if there are other issues that the patient would like to discuss, and the priorities of the patient. Personal goals are formulated in the third step, which can range from purely medical goals to social goals. In the fourth step, the healthcare provider and patient will together choose the right interventions to achieve these goals, based on the experience of the healthcare provider, the ideas of the patient, and a list of regional options. Different methods can be used to achieve these goals (i.e., through self-management, with e-health, with coaching from a non-medical care provider, with coaching from a healthcare provider within the GP practice, or with coaching from a healthcare provider outside the GP practice). The goals and interventions are documented in a personal healthcare plan, which is preferably digitally available to all relevant healthcare providers and the patient. Then, referrals are made if necessary, and the treatment is started. An evaluation is planned and carried out, if necessary multiple times. If a treatment goal is reached or another treatment goal is more urgent, the cycle can be repeated. The development of this PC-IC approach is described elsewhere in this issue [34].

### 2.2. Incentives in Payment Models

To design a payment model that would match the PC-IC programme of OPTIMA FORMA, we first studied the incentives for providers and other stakeholders that are present in the current Dutch healthcare system for all types of healthcare services used by patients with chronic disease. We classified these payment models according to the typology of Quinn (2015) [35] and identified the incentives related to these payment methods. Quinn (2015) [35] classifies eight basic payment methods in health care: (1) Per time period (budget/salary), (2) Per beneficiary (capitation), (3) Per recipient (contact capitation), (4) Per episode (case rates/per stay/bundled payments), (5) Per day (per diem/per visit), (6) Per service (fee for service (FFS)), (7) Per dollar of costs (cost reimbursement), and (8) Per dollar of charges (percentage of charges). 

Secondly, we studied incentives for stakeholders in innovative payment models. These innovative payment models were identified through the alternative payment model (APM) framework described by the Health Care Payment Learning and Action Network (HCP-LAN) [36]. The identified alternative payment models were: (1) pay for performance, (2) shared savings models, and (3) (sub)population-based bundled payment. We combined elements of these models to design an alternative payment model to stimulate PC-IC care for people with chronic diseases.

### 2.3. Design of an Alternative Payment Model

In the next step, we selected three alternative payment models and explicitly focused on the distinctive elements in their design. Since we aimed to propose an alternative payment model for the Dutch setting, the selection was based on two criteria, namely comprehensiveness and origin in the Dutch setting. The selection included:a population-based bundled payment model with an explicit incentive for quality of care of Cattel and Eijkenaar [37].a shared savings model of Hayen et al. [38].the alternative payment model of Steenhuis et al. [39].

We combined the design elements and design choices that were mentioned by these models into Table 1. Table 1 was used to guide the design of an alternative payment model that would fit the PC-IC programme OPTIMA FORMA. The design choices made were primarily informed by theory on provider-incentives and results from previous evaluation studies of the identified innovative payment models: (1) pay-for-performance [40,41], (2) shared-savings models [13,15,42,43], and (3) (sub)population—based bundled payments [37,44,45]. 

### 2.4. Expected Impact on Integration of Care

In the last step, we projected the expected impact of the innovative payment model on the integration of care, using the spider-web linked to the typology of Stokes et al [46]. This typology classifies the level of integrated care on eight domains: (1) Target population, (2) Time, (3) Sectors, (4) Provider coverage, (5) Financial pooling/sharing, (6) Income, (7) Multiple disease/needs focus, and (8) Quality measurements [46]. The higher the number, the higher the level of integration (1 = integration is poorly stimulated, 2 = integration is mediately stimulated, and 3 = integration is highly stimulated). 

## 3. Results

### 3.1. Incentives Induced by Different Payment Models

In Table 2, we provide a summary of current and alternative payment models to fund care for patients with chronic diseases in the Netherlands. 

Table 2 provides insight into the incentives induced by each payment model. None of the presented payment models above fully incentivises PC-IC. The SDMPs are currently funded by a fixed annual fee, which is paid in three monthly instalments (chronic care episode). The bundle primarily includes the GP, practice nurses, and a few paramedics working in the primary care sector. Hence, it stimulates collaboration between these service providers, but not beyond. It is likely to improve the quality and efficiency in primary care, but it also creates an incentive for adverse selection and referral of complex patients to secondary care. This is present, even though the fixed fee is based on a weighted average of resources used by patients with different severities. It also stimulates so-called ‘over-bundling’, referring to the incentive to enrol more patients than necessary. These undesired incentives can be mitigated by carefully combining elements of different payment models [37]. From a theoretical perspective, a bundled payment with a broader scope in terms of target population and services, in combination with a shared savings model and a pay-for-performance model seems promising [37].

### 3.2. Proposed Payment Model for Person-Centred and Integrated Care

Figure 2 shows the proposed payment model for all patients with one or more chronic conditions, starting with those that are currently included in the existing bundles for DM2, CVR, and COPD. The patient population is delineated by diagnosed chronic disease (at least DM2, CVR, or COPD), insurance (the patient has to be insured at one of the participating health insurers), and GP-practice (GP-practice has to collaborate with one of the participating primary care cooperatives). The payment model consists of three parts: (1) a person-centred bundled payment, (2) a shared savings model that pertains to all healthcare costs, and (3) a pay-for-performance part. 

Part one is a person-centred bundled payment that will be prospectively paid to the primary care cooperatives. For each patient, a personal healthcare plan is designed within the OPTIMA FORMA project (Figure 1). The services that can be included in the personal healthcare plan are shown in Figure 3. The bundled payment is based on the weighted average sum of all included services. The weighting is based on the number of patients that use a service and the costs of the service. The primary care cooperative is responsible for the coordination, organization, and financing of all subcontracted participating providers since the primary care cooperative is the main contractor. 

Part two is a virtual budget that contains all expected (healthcare) costs of these patients (the contracted bundled payment and the contracted expenditures outside the bundled payment). The case-mix adjusted weighted virtual budget will be compared to the realised expenditures to estimate the savings or losses. It is important to cap the expenditures, so the primary care cooperative does not bear the risk for patients with extreme high (unexpected) expenditures. One could start with a one-sided shared savings model, meaning that only the savings and not the losses will be shared between the health insurer and the primary care cooperative in the region, to mitigate risks for the primary care cooperative and avoid adverse behaviour. The savings will be distributed in a prespecified ratio between the primary care cooperative and the health insurer.

In part three, the prespecified ratio to share the savings depends on the quality of the delivered care. This pay-for-performance part depends on the measured performance of the monitored quality indicators. It is important to avoid time-consuming checklists and process indicators and adopt a small set of key outcome indicators. This requires trust from the health insurers and leads to more flexibility for providers to only provide services that are applicable for a patient instead of ticking boxes to show that they followed the correct process. Quality indicators are measured at primary care cooperative level. 

More details are provided in Appendix A.

The contract between the insurer and the primary care cooperatives should be signed for multiple years, preferably for three to five years. This provides the opportunity to explore the potentials of the alternative payment model (stimulate integration of care, improve quality of care, and reduce overall healthcare costs) and to gain mutual trust between the different stakeholders [39,43,75]. When the contract is renewed, changes can be made accordingly. For instance, after three or five years the one-sided shared savings model could be transformed into a two-sided shared savings model (the primary care cooperative also shares in the potential losses). A two-sided shared savings model stimulates cost-conscious behaviour better, but also increases the financial risk for the primary care cooperative [12,13,76]. 

### 3.3. Consequences of the Proposed Payment Model

The suggested alternative payment model is expected to be associated with incentives presented in Table 3. Each of the three parts of the proposed payment model has desirable and undesirable consequences, and the latter can be mitigated by the other part(s). 

As the range of services that can be included in the individual care plan (Figure 3) is much wider than in the current bundle for SDMP, the person-centred bundled payment is expected to stimulate the holistic approach that is aimed for by the PC-IC programme. The primary care cooperative and its associated care providers will have an incentive to improve efficiency by better coordination and collaboration because the budget extends over a wider range of services. This increases mutual responsibility. One of the perverse incentives of a bundled payment that may not cover the full care path of a patient, is that patients are referred to services outside the bundle [50]. The shared savings model mitigates this perverse incentive because the comparison of the actual and the expected expenditure (i.e., the virtual budget) pertains to the total healthcare expenditure. This could result in cost-conscious behaviour [38,77]. The current bundles for SDMPs do not incorporate a shared savings model. If the shared savings model stimulates cost savings through increased efforts to slow down the progression of disease and prevent acute hospital admissions, it also improves health outcomes. However, to mitigate financial risks for the primary care cooperative, a one-sided shared savings model is preferred over a two-sided shared savings model to avoid adverse behaviour of the primary care cooperative, especially at the beginning [78]. A perverse incentive of the person-centred bundled payment model and a shared savings model is cutting costs on necessary care. The pay-for-performance part of the model aims to reduce this risk by stimulating a high quality of care. 

Like all payment methods, this alternative payment model still induces some undesirable consequences which are hard to eliminate by one of the three parts of the payment model. The risk of reducing costs by cutting necessary care might be there to some extent. Furthermore, the threshold can be lowered to include patients in the person-centred bundled payment for whom one may expect little cost. However, adequate case-mix adjustment and capping costs could reduce these risks. To some extent, the person-centred bundled payment also reduces the choice of the patient because certain care providers are contracted, and others might not. As the personal healthcare plan is based on the needs, capabilities, and wishes of the patients, it is important that the contracted providers are able to provide the services shown in Figure 3 [39]. Another undesired consequence is that the primary care cooperative bears too much risk because all expenditures are included in the virtual budget. The primary care cooperative might not be able to control all of these expenditures. The incentives of providers outside the person-centred bundled payment are not well aligned because these physicians are mostly paid FFS. It is important that these providers feel motivated to collaborate. This might be achieved by investing part of the savings in joint quality improvement and innovation plans, which are attractive to these providers as well. Every pay-for-performance model introduces a risk for gaming behaviour, but the size of that risk depends on the proportion of a provider’s income that comes from the quality-payment. The challenge is to strike a balance between a sufficiently large proportion to incentivize quality improvement and a sufficiently small proportion to avoid gaming [77].

### 3.4. Impact on Integration

Figure 4 shows the degree of integration of the proposed payment model and the currently used bundled payments for the SDMPs on the eight dimensions of the framework based on Stokes et al [46]. Table 4 explains the levels that were expected for each domain.

## 4. Discussion

The aim of this paper was to design a bundled payment model that incentivises the transition from single-disease management to PC-IC for patients with chronic diseases. Based on a targeted literature review, we identified the incentives which are (theoretically) generated by the eight basic payment methods classified by Quinn [35] and the alternative payment models identified through the APM framework [36]. Based on the identified incentives, we designed an alternative payment model for PC-IC that consists of three main elements, i.e., (1) a person-centred bundled payment, (2) shared savings, and (3) pay-for-performance. The combination of these elements is expected to provide well-aligned, desired incentives towards multi-disciplinary collaboration to meet a patient’s needs, capabilities, and preferences. Each element is necessary to mitigate the undesired incentives of other elements. Furthermore, adequate risk-adjustment and cost-capping are prerequisites to mitigate large risks for providers and to mitigate adverse behaviour.

The implementation of this alternative payment model comes with certain challenges. The first challenge pertains to the investment of resources needed for implementation, which mainly include financial investments (e.g., transition costs to the alternative payment model) and time investments (e.g., to expand collaborations) [39]. To manage the alternative payment model, the software in place should be adapted to monitor the costs and quality of care over time [39]. Administrative costs of monitoring quality of care and negotiating about the conditions of the contract may increase, but this may be offset by a reduction in administrative costs when the services no longer have to be separately claimed [39]. 

The second challenge is to define the patient population that will be included in the person-centred bundled payment. The population of patients with DM2, CVR, and/or COPD is very heterogenous in terms of patient-characteristics, disease-severity, and co-existing morbidity patterns. For an adequate estimation of the expected expenditures, necessary to determine the savings or losses, clear inclusion and exclusion criteria need to be defined. 

The third challenge is to estimate an appropriate budget for the person-centred bundle. The budget will be estimated by a weighted sum of the costs of all (health)care modules provided in the bundle. The weighting will be carried out by predicting the number of patients that would use the various modules. As time after implementation progresses, figures regarding the relative use of the modules will become more reliable. Specifically, for OPTIMA FORMA, a clinical and economic evaluation study is planned that will provide the first estimates of the utilization of specific services. Micro-costing studies are necessary to determine the costs per module. 

For an appropriate comparison of expected and actual expenditures and to avoid extreme savings or catastrophic losses [79], adequate adjustment for differences in case-mix is important. Many countries with a multiple payer system (e.g., multiple social health insurers) like in the Netherlands, apply some form of risk equalization to distribute [part of] the budget among the payers. Whether variables included in the risk equalization formula of the health insurance system can also be used to adjust for differences in the case-mix of providers remains to be investigated. It is obvious that variables that are influenced by the PC-IC programme cannot be used in the case-mix adjustment because that would diminish/eliminate the estimated effects [38].

Another challenge when designing the alternative payment model is to determine the quality indicators for the pay-for-performance part of the alternative payment model. It is important to select quality indicators that are sensitive to improvements by the PC-IC programme and the alternative payment model. Based on a systematic literature review, specific design features that contribute to the desired effect of pay-for-performance are: (1) using outcome measures that are very specific and easy to track; (2) targeting individuals or small teams; (3) using absolute rather than relative targets; (4) frequently paying with little delay after delivery; and (5) involving providers from the start in the design [40]. Primary care cooperatives are reluctant to accept financial responsibility for indicators they cannot influence [80]. Conceptually, one would like to have one or more indicators for each of the four aims of PC-IC, but the challenge is to find the right balance between registration burden [79] and information need. 

To increase the chances of the successful implementation of PC-IC, several requirements need to be met. In their paper on the successful implementation of integrated care for people with multimorbidity, Looman et al [29] stressed the importance of ten mechanisms, of which one is securing long-term funding and adopting an innovative payment model that overcomes fragmentation. However, most important is constructive alignment, meaning that simultaneous measures at the micro, meso, and macro levels are needed to support the implementation of PC-IC [29]. With respect to the payment model, this implies that the incentives for all participating healthcare providers, as well as with existing financial streams, have to be aligned [39,80]. 

A more fundamental question that arises is whether a population-based payment model that would extend to the entire population in a geographically defined area (e.g., a region) and all care providers within that area would not be a more appropriate alternative compared to the alternative payment model proposed here. Especially, because that would stimulate prevention of disease and network care for the entire population in the catchment area, all of which is paid for from one bundled budget [50]. On one hand, it could fit the integrated nature of the PC-IC programme, but on the other hand, the step from the currently used bundled payments to a population-based payment might be too big. As it currently stands, the PC-IC programme OPTIMA FORMA focusses on people with the mentioned chronic diseases. If the population of interest were defined as the entire population of insured people in a region, the effect of the PC-IC programme could easily be diluted. That does not alter the fact the PC-IC programmes would benefit from economies of scale, which could reduce the financial risks for primary care cooperatives. 

## 5. Conclusions

To conclude, we designed a payment model with well-aligned incentives to support the adoption of PC-IC. This model consists of: (1) a person-centred bundled payment; (2) a shared savings model; and (3) a pay-for-performance part in which the sharing ratio between insurer and provider is conditional on the performance of the provider. This alternative model is likely to be an adequate alternative for the relatively limited bundled payment model that is currently used to fund the SDMPs in the Netherlands.

## Figures and Tables

**Figure 1 ijerph-20-03857-f001:**
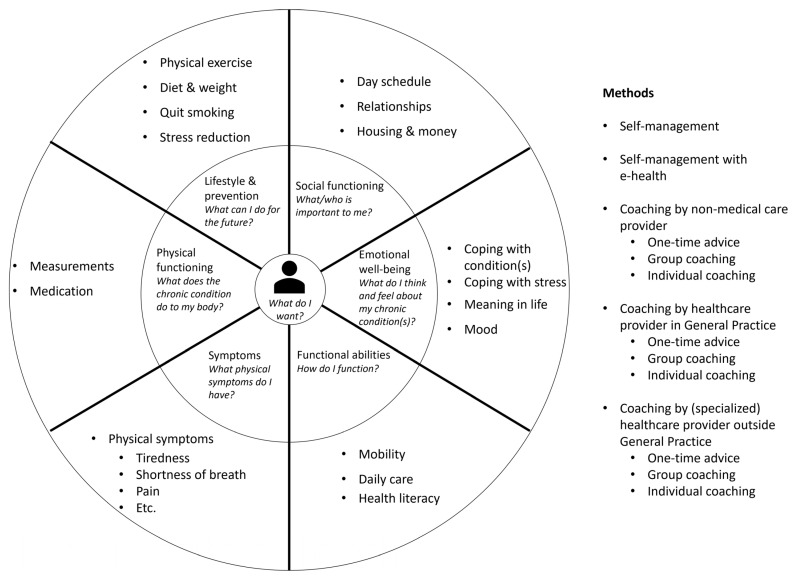
Person-centred and integrated care programme for people with chronic diseases.

**Figure 2 ijerph-20-03857-f002:**
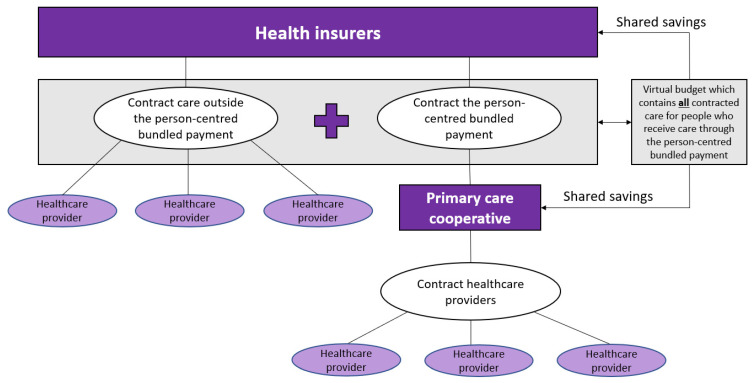
Conceptualization of the person-centred bundled payment in combination with the virtual budget and the shared savings.

**Figure 3 ijerph-20-03857-f003:**
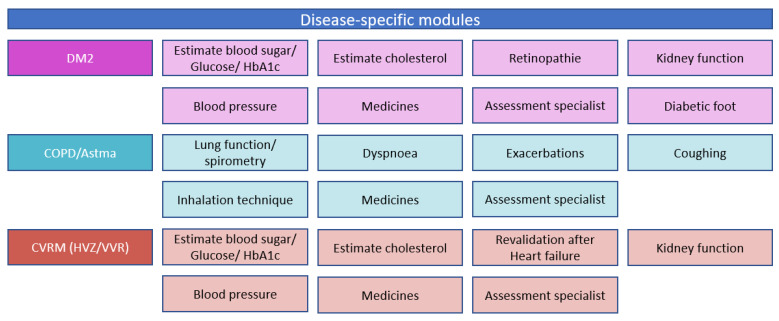
Disease-specific and disease-transcending modules provided through the new care plan and delivered if needed by the patient.

**Figure 4 ijerph-20-03857-f004:**
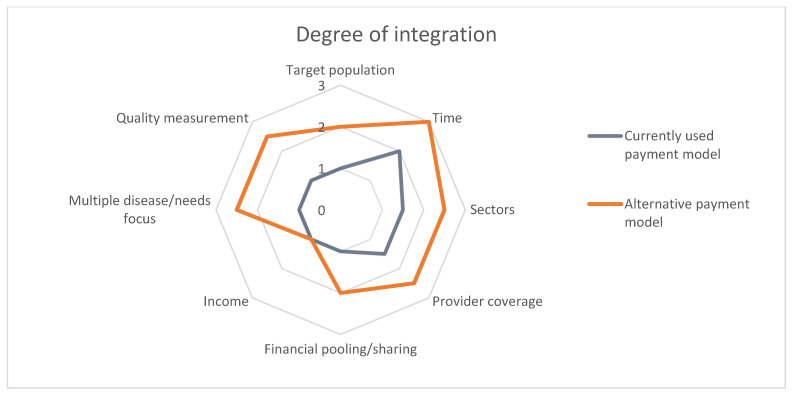
Degree of integration.

**Table 1 ijerph-20-03857-t001:** Design elements for the implementation of an alternative payment model.

	Design Elements
Specify bundle and select provider [s]: characteristics of a contract specified by a health insurer and one contracting entity or multiple healthcare providers	How to delineate the population?
	Definition of the patient population?
	Small or heterogeneous patient populations?
	How to attribute patients to a provider group?
	Which providers are included?
	Mandatory or voluntary bundled payment?
	Who is the main contractor?
	Are the group members employed or subcontracted?
	What care services are included in the bundle?
	[Re]Allocation of care delivery among providers?
	Prospective or retrospective payment strategy?
Negotiate and sign contract: negotiate about price, volume, weight of case-mix method, quality measurement and quality incentive structure, distribution of savings/losses, and risk-mitigating measures for providers	Is the payment real or virtual?
	How to set a payment/target? (Calculate the average annualizing expenditures, Weight the expenditures, and Cap expenditures, evaluate expenditures against a benchmark, Trending factor, Risk adjustment)
	Allocation of possible savings?
	One-sided or two-sided risk?
	What is the risk-sharing rate?
	Is there a maximum saving rate according to the costs?
	Is risk adjustment applied?
	Which risk adjustors are used?
	What is the contract duration?
	What care to carve out?
	Are shared savings/losses conditional on quality?
	Add-on for quality?
	Which quality indicators to use?
	What measurement level [individual/group]?
	Rewards and/or penalties?
	Maximum payment size relative to total payment?
	Absolute, relative and/or improvement targets?
	How often to pay for performance?

**Table 2 ijerph-20-03857-t002:** Current Dutch payment models based on the typology of Quinn (2015) [35] and the HCP-LAN (2017) [36] and their (financial) incentives.

Payment Model	Providers	Incentivises	Literature
**Combination of contract capitation payment and fee for service**	General practitioner	***Positive*** Fewer referrals Adequate provision of care (no under- or overtreatment) Preventive care ***Negative*** Collaboration not explicitly addressed	[20,47,48,49]
**Fee for service (FFS)**	Physiotherapist, exercise therapist, speech therapist, dietician, and district nursing	***Positive*** High productivity Transparency of the delivered care ***Negative*** Overprovision Unnecessary readmissions and diagnostics Quality of care not explicitly addressed Fragmentation of care Preventive care not stimulated	[50,51,52,53]
**Per time period**	Salary paid health care providers	***Positive*** Cost-conscious behaviour ***Negative*** Undertreatment Selection of patients Low productivity which could result in longer waiting lists Quality of care not explicitly addressed Unnecessary referrals	[54,55,56]
**Per episode** (chronic care)	The SDMPs for chronic diseases in primary care	***Positive*** Integrated care (coordination and continuity) Cost-conscious behaviour High quality of care Efficient care ***Negative*** Risk selection ‘Over-bundling’ * Double billing Unnecessary referrals Undertreatment within the bundle	[14,50,57,58,59,60]
**Per episode** (1. DTCs in specialist care, 2. Per patient profile and time, and 3. Per treatment activity and time)	Medical specialists General basic psychologists Specialist psychologists	***Positive*** Less fragmentation of care Efficiency ***Negative*** Strategic declaration behaviour Overprovision Upcoding	[61,62,63]
**Contract capitation in combination with a patient co-payment** (€19 per month)	Social care	***Positive*** Wide access because of the low co-payment Flexibility to tailor help to personal circumstances ***Negative*** Underuse because of accumulating co-payments	[64,65]
**Pay for performance**	Alternative payment model	***Positive*** Quality of care explicitly addressed Transparency of care Positive spill-over effects ** ***Negative*** Negative spill-over effects ** Risk selection Gaming behaviour No incentive if threshold is met Providers held accountable for outcomes they may not be able to influence	[37,40,41,66,67,68,69,70]
**Shared savings model**	Alternative payment model	***Positive*** Cost-conscious behaviour Integrated care (coordination and continuity) Preventive care Quality of care (implicitly and explicitly) ***Negative*** Undertreatment To drive the expenditures of the benchmark	[13,14,15,42,43,71,72,73,74]
**(Sub)population—based bundled payment**	Alternative payment model	***Positive*** Integrated care (coordination and continuity) Quality of care Cost-conscious behaviour Preventive care ***Negative*** High risk for the contracting entity Reduction of necessary care in the bundle Less freedom of choice Risk selection	[16,17,18,45]

Note: DMPs = disease management programmes, DTCs = disease-treatment combinations. * Over-bundling refers to the incentive to broaden the diagnoses, as to increase the target population for which a provider receives a bundled payment (e.g., including pre-diabetes patients at risk of developing diabetes in the bundled payment of diabetes patients). ** Positive spill-over effects occur when non-measured outcomes improve as well, as a consequence of the extra attention for the measured outcomes. Negative spill-over effects occur when a focus on measured outcomes leads to non-measured outcomes being neglected and deteriorating.

**Table 3 ijerph-20-03857-t003:** Consequences which are generated by the proposed alternative payment model.

Payment Model	Desirable Consequences	Undesirable Consequences
Person-centred bundled payment	A holistic approach Integration of care More flexibility on how to spend the budget Responsibility of the primary care cooperative and therefore coordination of care Reduction of risk selection	Lowering the threshold to include someone in the person-centred bundled payment Reduction of freedom of choice of the patients because certain physicians are contracted, and others are not Reduction of costs by avoiding necessary care
One-sided shared savings	A holistic approach Multidisciplinary collaboration due to mutual responsibility The right care for the right patient at the right place Cost-conscious behaviour Double declaration is unattractive Mitigated risks for the primary care cooperative	Feeling less responsible because the savings partly depend on providers that are not part of the person-centred bundled payment, which makes the coordination difficult Reduction of costs by avoiding necessary care
Pay-for-performance	High quality of care	Focus on the measured quality indicators (gaming)

**Table 4 ijerph-20-03857-t004:** Description of the level of integration expected to result from the proposed payment model.

Domain (Level)	Explanation
**Target population** (2)	The currently used payment model only focusses on people with either DM2, COPD, or CVRM. The alternative payment model includes care for all three chronic diseases in one bundle and additional disease-transcending care. The alternative payment model focuses on a much wider population; therefore, the level of integration moves from 1 to 2.
**Time** (3)	In the currently used payment model, agreements about the budget are made for one year. We recommend making agreements for multiple years. The collaboration becomes stronger and the mutual trust between primary care cooperative and health insurer increases. Agreements for a longer time span also create more possibilities to innovate and investigate the potentials of the alternative payment model; therefore, the level of integration moves from 2 to 3.
**Sectors** (2.5)	The currently used payment model focusses on primary care. Secondary care is incorporated, but is rather limited and only includes a single consultation by a specialist for a small proportion of the target population. The aim of the alternative payment model is to finance care from all sectors (primary, secondary, tertiary care, and the social domain). The primary care cooperative is a coordinating organ, so also non-financial agreements could be made with, for instance, the municipality about the social care domain. Both the SDMP and the PC-IC programme include preventive interventions like smoking cessation support and lifestyle interventions, but so far these have not been covered by the bundled payment for SDMP. Therefore, the level of integration moves from 1.5 to 2.5.
**Provider coverage** (2.5)	The providers covered by the currently used payment model are the practical nurse, the GP, the dietician, the foot therapist, the physiotherapist, and a consultation with a medical specialist. In the alternative bundled payment, we propose to expand the scope to include all services that are part of the personal care plan (Figure 3). All other healthcare utilization is included in the virtual budget (Figure 2); therefore, the level of integration moves from 1.5 to 2.5.
**Financial pooling/sharing** (2)	In the currently used payment model, the primary care cooperative and health insurer do not usually have agreements about sharing savings or losses. These savings or losses are estimated by comparing a virtual budget (i.e., the expected expenditures) to the real expenditures. We advise to start with a one-sided shared savings model, and therefore, the level of integration moves from 1 to 2.
**Income** (1)	The alternative payment model for PC-IC will not drastically change the income of the individual health care provider. The budget for chronic care of the GP practice increases as the target population increases, but at the same time less care will be financed through FFS. The net result depends on the details of the contract.
**Multiple diseases/needs focus** (2.5)	The currently used payment model finances disease-specific care. The alternative payment model includes all services that are part of the personal care plan (Figure 3). At the start, the model will pertain to people with DM2, CVRM, and/or COPD, but once a patient is incorporated into the PC-IC programme, the patient will be fully assessed on six domains (Figure 1). Therefore, the level of integration moves from 1 to 2.5.
**Quality measurement** (2.5)	In the currently used SDMP, the quality of care is assessed by InEeN, which delivers an annual report about the quality of chronic care. The quality indicators are determined by the Dutch GP society (NHG) and mostly include process indicators (e.g., if the smoking status is registered). The currently used bundled payment is not related to performance on these indicators. In the alternative payment model, we aim to measure quality of care on outcome indicators (e.g., health-related quality of life) and patient satisfaction. The ratio that is used to share savings between the health insurer and the primary care cooperative will depend on the delivered quality of care. Therefore, the level of integration moves from 1 to 2.5.

## Data Availability

Data sharing not applicable.

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
