# Peer review of "How Can a Bundled Payment Model Incentivize the Transition from Single-Disease Management to Person-Centred and Integrated Care for Chronic Diseases in the Netherlands?"

_ijerph, 2023, doi:10.3390/ijerph20053857_

Round 1
Reviewer 1 Report
General comments:
Thanks for the opportunity to review this manuscript. I do research on health policy, but do not specialize in payment design and provisions, and certainly my familiarity with the Dutch health system is limited. As a result, I’ve tried to provide a few general comments for the authors to think about, while trying to point out places that easy fixes can be made to improve the manuscript’s clarity and organization. In general, I found this manuscript to be a readable and thoughtful exploration of how to adjust payment models to maximize positive incentives while minimizing negative ones.
Things I like.
1. I like how this manuscript is trying to blend different types of existing payment models.
2. I am not particularly familiar with the details of payment plans, and I found Table 3 and the theory behind it of how to combine elements of payments systems to try to maximize good incentives and eliminate bad ones extremely helpful and illuminating. There is a lot going on in these tables, but each element is straightforward enough that it makes them digestible. There's a comprehensive attempt to build theory here that is missing in many medical and even health policy journals.
Things to work on:
1. One thing I find missing from the discussion is the administrative costs of negotiating and implementing these sorts of hybrid payment agreements, which seem to me to be likely much more complicated than existed bundled payment or fee-for-service arrangements. Are the benefits of this model outweighed by the increase in administrative costs? Or do you disagree with my premise that administrative costs will rise?
2. You say that most of the challenges of your model aren’t country specific, but why? A bit of discussion would be helpful.
Small corrections and language things:
NOTE: I speak and write American-style English, so please ignore any of my suggestions that conflict with the British-style English that seems to be the standard for this journal. Overall, the standards of fluency in this article are extremely high. I only found minor things lost in translation, none of which were central to meaning.
1. Page 1: GP should be written out on first reference (it is written out further down the page)
2. Page 2: “Two health insurers” in Germany. Do those insurers cover most of the population or are they small, public or private? It would be nice to have context.
3. Page 2 “ensured by two health insurers” should be “insured”
4. Page 2: “Lumpsum” should probably be two words.
5. Page 2: I’m sure the term is Dutch, but writing out the words in the acronym “INEEN” on first reference would be helpful.
6. Page 3: first full paragraph: “avoid the costs to exceed the bundled payment” should be “avoid costs exceeding the bundled payment”
7. Page 3: Last full paragraph: “The case manager discusses if the results are recognizable…” I don’t think “recognizable” is the correct word to convey your meaning here. Perhaps “The case manager ensures the patient understands the results” represents your intent?
8. Page 6: Should Section 3.1 be labeled “subsection?” or did you have a more specific idea in mind?
9. Table 2: In the “Per Time-Period” Section It’s unclear what “High Quality of Care not explicitly” is referring to.
10. Note for Table 2: Should ‘indication’ be ‘diagnoses? (This might be an American-British English thing)
11. Page 8, first paragraph: no comma in “Part one, is…”
12. Page 9: First paragraph after Figure 3: Try ‘preferably” instead of ‘desirable’
13. Page 12: Last paragraph “Cutting on necessary care” should be “cutting necessary care”
14. Page 13 second paragraph “multi-morbidity” should probably be written as “multiple morbidities”
15. There are many places where economies of language can be made by eliminating unnecessary adjectives or even verbs. For example, on page 2 cutting the word “rather” from “rather limited” in the third full paragraph saves a word without altering meaning. Another one on Page 2 is in the last paragraph: “chronic care programmes are focusing on a single chronic disease” could become “chronic care programmes focus on a single disease” One example on page 3 in the second full paragraph reads “This paper aims to present” could simply read “This paper presents” Each of these examples only save 1-2 words, but they improve readability and add up over time to a significant number of words that could be used to further discuss the central points of the manuscript in a bit more depth.
Author Response
Dear Reviewer,
Thank you for reviewing our manuscript. We tried to respond to all comments. In the revised manuscript you can see all the changes that we have made.
We tried to comment on the administration costs in the discussion.
The other comments mainly focused on language corrections. You can find the corrections in the revised manuscript.

Reviewer 2 Report
This is a really strong paper on the design and implementation of a novel person-centered and integrated care bundled payment model in the Netherlands. The article was coherent, and the content was very pertinent.
I suggest minor changes to improve the manuscript.
Introduction
· Page 2/21
o “Nowadays, there are 130 primary care cooperatives in the Netherlands, primary being collaborations of general practices” to ….Netherlands based on the collaborations of general practices.
o First use of the acronym INEEN – provide the full name.
Methods
· Page 3/21
o “Interventions to establish these goals are put into place” : change establish to achieve.
o “different methods can be used to reach these goals” : change reach to achieve.
Results
· Page 6/21
o Table 2 – column 3: Remove the … after Incentivises
o Table 2: You provide positive and negative features of all payment models except for the combination of contract capitation payment and fees for services – are we to understand that there are no negative elements of this model?
Author Response
Dear Reviewer,
Thank you for reviewing our manuscript. In response to your comments we changed table 2.
We also corrected the textual errors you pointed out. You can find those in the uploaded revised manuscript.
